# TRPC Channels: Dysregulation and Ca^2+^ Mishandling in Ischemic Heart Disease

**DOI:** 10.3390/cells9010173

**Published:** 2020-01-10

**Authors:** Débora Falcón, Isabel Galeano-Otero, Marta Martín-Bórnez, María Fernández-Velasco, Isabel Gallardo-Castillo, Juan A. Rosado, Antonio Ordóñez, Tarik Smani

**Affiliations:** 1Department of Medical Physiology and Biophysics, Institute of Biomedicine of Seville, University of Seville, 41013 Seville, Spain; dfalboy@gmail.com (D.F.); igaleano@us.es (I.G.-O.); mmartin55@us.es (M.M.-B.); 2Biomedical Research Networking Centers of Cardiovascular Diseases (CIBERCV), 28029 Madrid, Spain; mvelasco@iib.uam.es; 3IdiPAZ Institute for Health Research La PAZ, 28029 Madrid, Spain; 4Department of Stomatology, School of Dentistry, University of Seville, 41009 Seville, Spain; igallardo@us.es; 5Department of Physiology (Cell Physiology Research Group), Institute of Molecular Pathology Biomarkers, University of Extremadura, 10003 Caceres, Spain; jarosado@unex.es

**Keywords:** TRPC channel, Ca^2+^ entry, cardiac infarction, cardiac repair

## Abstract

Transient receptor potential canonical (TRPC) channels are ubiquitously expressed in excitable and non-excitable cardiac cells where they sense and respond to a wide variety of physical and chemical stimuli. As other TRP channels, TRPC channels may form homo or heterotetrameric ion channels, and they can associate with other membrane receptors and ion channels to regulate intracellular calcium concentration. Dysfunctions of TRPC channels are involved in many types of cardiovascular diseases. Significant increase in the expression of different TRPC isoforms was observed in different animal models of heart infarcts and in vitro experimental models of ischemia and reperfusion. TRPC channel-mediated increase of the intracellular Ca^2+^ concentration seems to be required for the activation of the signaling pathway that plays minor roles in the healthy heart, but they are more relevant for cardiac responses to ischemia, such as the activation of different factors of transcription and cardiac hypertrophy, fibrosis, and angiogenesis. In this review, we highlight the current knowledge regarding TRPC implication in different cellular processes related to ischemia and reperfusion and to heart infarction.

## 1. Introduction

The heart rate of a healthy adult ranges between 60 and 100 beats/min, which is mainly achieved by adequate function of the cardiac contraction/relaxation cycle. Adequate ventricular contraction is strongly dependent on effective excitation–contraction (EC) coupling in cardiac cells. Electrical stimuli travel across conducting cardiac tissues to the cardiomyocytes, inducing a cell-membrane depolarization activating ion channel and finally activating the cell contractile machinery (reviewed elsewhere [1,2]). EC coupling and cell contraction are critically dependent on Ca^2+^ influx and Ca^2+^ channel trafficking. The initial cell-membrane depolarization stimulates sarcolemma L-type Ca^2+^ channels, prompting a small influx of Ca^2+^ from the extracellular medium. Ca^2+^ entry triggers a large release of Ca^2+^ from the sarcoplasmic reticulum via ryanodine receptors (RyR), resulting in an increase in the intracellular Ca^2+^ concentration ([Ca^2+^]_i_). The rise in [Ca^2+^]_i_ boosts Ca^2+^ binding to troponin C, which activates the contractile machinery. After contraction, [Ca^2+^]_i_ must decrease to allow cell relaxation, which is achieved mainly via two mechanisms: Ca^2+^ re-uptake by the sarco-endoplasmic reticulum Ca^2+^ ATPase (SERCA) pump and Ca^2+^ efflux by the sarcoplasmic Na^+^/Ca^2+^ exchanger (NCX) [2,3]. Dysregulation of any of these Ca^2+^ handling processes is commonly associated with cardiac dysfunction.

Recently, other players emerged as key partners in the regulation of cardiac Ca^2+^ handling. Among these partners are the transient receptor potential (TRP) channels that are classified in a superfamily, including 28 mammalian TRP proteins divided according their genetic and functional homology into six families: TRPP (polycystin), TRPV (vanilloid), TRPM (melastatin), TRPA (ankyrin), TRPML (mucolipin), and TRPC (canonical). TRP channels are composed of six transmembrane domains (TM1–TM6), with a preserved sequence called the “TRP domain” adjacent to the C-terminus of TM6 and a cation-permeable pore region formed by a loop between TM5 and TM6 (reviewed in Reference [4]). TRP channels are located in the plasma membrane, and their activation allows the entry of Ca^2+^ and/or Na^+^, with higher permeability for Ca^2+^. Although most TRP channels lack a voltage sensor, they can be activated by physical or biochemical changes, regulating Ca^2+^ dynamics by directly conducting Ca^2+^ or prompting Ca^2+^ entry secondary to membrane depolarization and modulation of voltage-gated Ca^2+^ channels [5]. The activation of different isoforms of TRP is associated with cell-membrane depolarization, for example, in smooth muscle cells [6,7] and in cardiac cells [8,9,10]. 

There is substantial evidence that TRP channels have important roles in mediating cardiac pathological processes, including cardiac hypertrophy and fibrosis [11,12,13], which all lead to deleterious cardiac remodeling and subsequent heart failure (HF). This review focuses on the role of TRPC channels and provides an overview of the most relevant and recent findings related to these channels and ischemia-related disease in the heart. Nevertheless, the activation mechanism of TRPC channels is not yet completely clarified, and even less so in cardiac cells. Previous studies using different cell types suggest that TRPCs can interact physically with different splice variants of the inositol triphosphate receptors (IP3R). For instance, TRPC1 [14], TRPC3 [15,16], and a splice variant of human TRPC4 [17] interact physically with the IP3R. Actually, it appears that IP3R and Ca^2+^/calmodulin compete for a common binding site on TRPC3 since the displacement of calmodulin by IP3R from the binding domain activates TRPC3 [18]. Others researchers proved that phosphatidylinositol 4,5-bisphosphate (PIP2) participates in the regulation of TRPC4 and TRPC5 [19,20]. Gα_q_ protein also activates TRPC1/4 and TRPC1/5 through direct interaction [21]. Meanwhile, independent studies demonstrated that TRPC3, 6, and 7 are activated by diacylglycerol (DAG) [22,23,24,25]. Interestingly, TRPC4 and 5 channels also become sensitive to DAG when their interactions with other regulators are inhibited, such as protein kinase C (PKC) and Na^+^/H^+^ exchanger regulatory factor (NHERF) [26]. 

## 2. TRPC Channels in the Cardiovascular System

TRPC channels are classified into seven members (TRPC1–7) that are distributed based on biochemical and functional similarities into TRPC1/4/5, TRPC3/6/7, and TRPC2, which is a pseudogene in humans. The expression of TRPC isoforms in the heart was examined in different stages of animal development, animal models, and areas of the heart. They are expressed at very low levels in normal adult cardiac myocytes but their expression and activity might increase in pathological processes [12,13,27]. However, they likely display different patterns of expression in cardiac cells isolated from the sinoatrial node and in myocytes isolated from atrial or ventricular heart [22,28]. In human cardiac tissues and/or neonatal rat cardiomyocytes, messenger RNA (mRNA) of TRPC5 [29,30] and TRPC6 [31] was detected. In animal models, the expression of TRPC1/3–7 was confirmed in adult rat and mouse ventricle and atrial cardiac myocytes either at mRNA or protein levels [13,32,33]. Other reports showed that TRPC1/3–6 are expressed in rat ventricular myocytes of fetal and neonatal ventricular myocytes [28,34]. In sinoatrial node cells, TRPC1, 2, 3, 4, 6, and 7 mRNA expression levels are detected using RT-qPCR, whereas TRPC5 expression is not observed. Furthermore, experiments using immunohistochemistry confirmed protein expression of TRPC1, 3, 4, and 6, but not TRPC7 in mouse sinoatrial node and in isolated pacemaker cells [35]. In the case of cardiac fibroblasts, all TRPC isoforms were described. In particular, the mRNAs of TRPC1, 3, 4, and 6 are detected in mouse cardiac fibroblasts. Meanwhile, isolated rat ventricular fibroblasts have significant mRNA expression of TRPC2, 3, and 5 [36,37,38]. Experiments using immunocytochemistry and Western blot also revealed the expression of TRPC1, 3, 4, and 6 proteins in rat and human cardiac fibroblasts [39,40,41].

A functional TRPC channel is composed of four proteins, allowing it to form homo or heterotetramers [42]. However, the concept of TRPC multimerization was barely addressed in cardiac myocytes. A previous study from Molkentin’s group suggested multimerization of TRPC3 and homotypic TRPC6 in adult mouse cardiac myocytes since they demonstrated, using an immunoprecipitation approach, that TRPC3 can associate with TRPC4 protein [5]. More recently, TRPC6 was suggested to form a heteromeric complex with TRPC3 and nicotinamide adenine dinucleotide phosphate hydrogen (NADPH) oxidase 2 (NOX2) protein in diabetic mouse heart. Nonetheless, this study used HEK293 cells to confirm the interaction between TRPC3 and TRPC6 by immunoprecipitation [43]. It should be noted that other studies indicated that TRPC channels can form a macromolecule complex with the NCX [44], Na^+^/K^+^ pump [45], and SERCA pump [46]. Therefore, they might create a microenvironment facilitating the fine-tuning of Ca^2+^ homeostasis and excitation–contraction coupling (reviewed elsewhere [47,48,49]). In fact, recent evidence confirmed that TRPC3 mediates Ca^2+^ and Na^+^ entry in proximity of NCX, elevating Ca^2+^ levels and cardiac contractility [44]. Certainly, more precise investigations about TRPC heteromerization will be welcome to reveal whether this concept is similar to that observed in other cells such as smooth muscle cell [50], platelets [51], hippocampus [52], or rat brain [53]. Actually, Bröker-Lai J et al. [52] combined quantitative high-resolution mass spectrometry with affinity purifications using isoform-specific antibodies on membrane fractions prepared from wild-type (WT) and target-knockout (KO) brains to demonstrate that TRPC1, 4, and 5 form heteromeric complexes in the brain, particularly in the hippocampus. 

## 3. TRPC Channels Mediate Ca^2+^ Influx in Cardiac Myocytes

There are considerable indications that, in cardiac myocytes isolated from the atrium, the ventricle, or from neonatal rat ventricular myocytes (NRVM), TRPC channels participate both in store-operated Ca^2+^ entry (SOCE) and receptor-operated Ca^2+^ entry (ROCE) pathways, and their activation and/or upregulation is essential for cardiac Ca^2+^ signaling, particularly under pathological situations (reviewed elsewhere [54,55,56]). Independent studies showed that DAG, which works as an important mediator of the G-protein coupled receptor (GPCR)-stimulated Ca^2+^ signaling pathway, activates TRPC3 and 6. For instance, Onohara et al. [10] demonstrated that stimulation of NRVM with angiotensin-II and 1-oleoyl-2-acetyl-*sn*-glycerol (OAG), a membrane-permeable DAG analogue, activates TRPC3 and 6 channels, causing membrane depolarization. They further demonstrated that small interfering RNA (siRNA) against TRPC3 and 6 significantly reduces responses to angiotensin-II. OAG also activates a cation current in mouse cardiac myocytes that is significantly reduced by cell dialysis with an anti-TRPC3 antibody [57]. Moreover, the activation of A1 adenosine receptor in atrial and ventricular myocytes activates TRPC3, through DAG, since Ca^2+^ influx is inhibited by Pyr3, considered a specific inhibitor of TRPC3 [33].

It should be noted that other studies focused on the role of TRPC channels in SOCE activation in cardiac myocytes. For instance, a recent study by Wen et al. [58] demonstrated the presence of SOCE in normal adult mouse ventricular myocytes and the participation of TRPC1, 3, and 6, since antibodies against these TRPC channels reduced store depletion-mediated Ca^2+^ entry. Previously, Wu et al. [5] characterized the participation of TRPC3, 4, and 6 in the exacerbated SOCE observed in mouse cardiac myocytes from hypertrophic hearts. They demonstrated significant reduction of SOCE mediated by specific inhibition of SERCA with cyclopiazonic acid in cardiac-specific transgenic mice expressing dominant-negative (dn) TRPC3 (dn-TRPC3), dn-TRPC6, or dn-TRPC4. The participation of TRPC3 and 4 in SOCE was also characterized in adult rat ventricular myocytes induced by specific activation of EPAC (exchange protein directly activated by cyclic adenosine monophosphate (cAMP)) with 8-(4-Chlorophenylthio) (8-pCPT) [12]. This study revealed significant upregulation of TRPC3 and 4, which correlates with an SOCE increase in 8-CPT-treated cardiac myocyte. In addition, thapsigargin-induced SOCE is inhibited by Pyr3, a TRPC3 inhibitor [12]. Another study suggested a role of TRPC1, 4, and 5 in SOCE caused by aldosterone stimulation of NRVM. Indeed, thapsigargin-induced SOCE is inhibited in aldosterone-treated NRVM transfected with dn-TRPC1 and dn-TRPC4, and with siRNA against TRPC5, whereas dn-TRPC3 did not alter SOCE [59]. Moreover, TRPC1 and 4 overexpression correlates with calcium release activated channel (CRAC)-like current recorded in isolated hypertrophied right ventricular myocytes treated with monocrotaline [60]. Likewise, we proposed that at least TRPC5 may be critical in SOCE since its downregulation inhibits thapsigargin-induced potentiated SOCE in NRVM under ischemia and reperfusion. We further demonstrated that TRPC5 colocalizes with Orai1, the pore-forming sub-unit of store-operated Ca^2+^ channel (SOCC) [13]. More recently, Bartoli et al. [61] proposed that TRPC1 and 5 are involved in aldosterone activation of SOCE in adult rat ventricular cardiomyocytes. This study revealed that cardiac myocytes treated for 24 h with aldosterone enhance SOCE through the activation of mineralocorticoid receptor, and increase the store-operated Ca^2+^ current (I_SOC_), which correlates with specific overexpression of TRPC1 and 5, as well as stromal interaction molecule 1 (STIM1), but not of TRPC3, 4, or 6, nor of Orai1 and Orai3. 

It is important to note that all these reports used agents that selectively deplete sarcoplasmic reticulum Ca^2+^ stores (e.g., cyclopiazonic acid, thapsigargin) to activate SOCC and avoid contribution of ROCE pathways. The combination of using different TRPC inhibitors, together with functional pore inhibitory antibodies for TRPC proteins and RNA silencing, suggests that TRPC channels account for the prominent SOCE in cardiac myocytes, especially under pathological conditions. Nevertheless and despite the increasing number of studies investigating SOCE in cardiac myocytes, the role of TRPC channels in SOCE is still controversial, which requires further investigation.

## 4. Role of TRPC Channels in Cardiac Pathophysiology

There is a general consensus that the overexpression and activation of TRPC channels are associated with deleterious cardiac pathology. As reviewed recently, under physiological conditions, the function of TRPC channels in the heart does not seem to be essential [4,62]. It appears that hearts from KO mice of different TRPC channels do not present any significant contractile abnormalities. Echocardiography analysis showed that TRPC3 KO and TRPC6 KO mice have similar resting left-ventricular mass and fractional shortening as compared to their respective littermate controls [63]. However, the induced stress-stimulated contractility, known as the Anrep effect, is diminished in isolated papillary muscles and cardiomyocytes from TRPC6 KO, but not TRPC3 KO mice [64]. In addition, TRPC1/4 double-KO mice have normal basal cardiac contractility, as well as normal systolic and diastolic functions. In contrast, isoproterenol-induced chronotropic responses are reduced in TRPC1/4 double-KO mice [65]. 

TRPC channels might play a role in some physiological processes. TRPC channels likely regulate cardiac pacemaking, conduction, ventricular activity, and contractility during cardiogenesis, through the interaction with the Ca_v_1.2 channel in isolated hearts obtained from four-day-old chick embryos [22]. TRPC channels also contribute to Ca^2+^ homeostasis by directly conducting Ca^2+^ or indirectly via membrane depolarization and voltage-gated Ca^2+^ channel modulation. The resulting TRPC-mediated Ca^2+^ influx is required for the activation of signaling pathways that play minor roles in the healthy heart. For instance, they are involved in the activation of transcription factors promoting cardiac hypertrophy, fibrosis, and/or arrythmia [5,13,28,55,66,67,68,69]. Here, we discuss the role of TRPC channels in processes related to cardiac ischemic diseases. 

## 5. Role of TRPC Channels in Cardiac Ischemia

### 5.1. TRPC Channels in Myocardial Infarction

One of the first pieces of evidence of the participation of TRPC in myocardial infarction (MI) was proposed using bioinformatic analysis combined with experimental approaches. Zhou et al. [70] demonstrated an increase in the expression of TRPC6, which was experimentally validated in a one-month post-MI rat model, suggesting TRPC6 as a potential therapeutic target for MI. Later, other studies highlighted the induction of TRPC proteins under MI and explored the idea that Ca^2+^ influx through TRPC channels overexpressed after MI contributes to cardiac dysfunction and adverse remodeling. In fact, significant increases in TRPC1, 3, 4, and 6 mRNA levels in mice one, two, and six weeks post MI were observed, as compared with sham [71]. This channel upregulation correlates with the increase in Ca^2+^ entry when myocytes isolated from MI adult mouse are stimulated with cyclopiazonic acid and OAG. Furthermore, mice expressing dn-TRPC4 have less pathological hypertrophy, better cardiac hemodynamic performance, and increased survival after MI, as compared with WT mice [71]. Therefore, the loss of TRPC4 function likely protects against the progression of cardiac dysfunction after MI. Interestingly, Jung et al. [72] suggested that gain of function of TRPC4 due to a genetic variation (I957V) causes an increase in channel activity, which has a protective effect against MI. The authors identified a single-nucleotide polymorphism (SNP) in TRPC4 that associates with MI risk in a case–control study. They further used multivariate analysis to show a protective effect of the I957V allele against MI risk, but only in diabetic patients. Therefore, the mutated TRPC4-I957V is thought to mediate higher Ca^2+^ signals, perhaps to facilitate the generation of endothelium and nitric oxide-dependent vasorelaxation. Nevertheless, the authors did not experimentally test this hypothesis. Recently, we observed significant dysregulation in the expression of several TRPC isoforms in a Wistar rat model of MI induced by transient ligation of the left coronary artery. A PCR-based micro-array, qRT-PCR, and Western blotting demonstrated significant upregulation of TRPC1, 3, 4, 5, and 6, whether in at-risk or in remote zones of infarcted hearts, as compared to sham. Specific downregulation of TRPC5 in MI rats infused with urocortin-2 at the onset of reperfusion was observed, offering a role of TRPC5 in cardioprotection [13]. 

In the case of TRPC3 and 6, a previous study determined that TRPC6 KO mice had significantly higher rates of mortality due to ventricular wall rupture throughout 3–7 days post MI [37]. In contrast, TRPC3/6/7 triple-KO mice subjected to transient MI (30 min of ischemia followed by 24 h reperfusion) exhibit reduced infarct size, better cardiac performance, and less cardiac tissue damage post MI, as compared with WT animals. In addition, they have reduced apoptosis through the inhibition of the calcineurin–nuclear factor of activated T cells (NFAT) signaling pathway [24]. These results suggest that TRPC3, 6, and 7 contribute significantly to worsening MI impacts on cardiac function. Further investigations will be welcome to clarify the discrepancy between these KO studies. It will be interesting to examine whether the cardioprotective effects observed in the triple-KO mice affect the transformation of myofibroblasts required during wound healing and scar formation. 

### 5.2. TRPC Channel Role in Ischemia and Reperfusion Injuries and Cardioprotection 

Ischemia and reperfusion (I/R) injury is the main cause of cell apoptosis and necrosis observed after an MI. Several studies demonstrated evidence linking cytosolic Ca^2+^ increase through TRPC and apoptosis after I/R [24,73]. Studies using TRPC inhibitors examined their role in I/R injuries. For instance, Kojima et al. [74] showed, in a Langendorff-perfused mouse heart under I/R, that left-ventricular functions are significantly improved by the administration of ion channel blockers (2-aminoethoxydiphenyl borate (APB) and La^3+^) during the initial 5 min of reperfusion, suggesting a TRPC channel role in contractile dysfunction in reperfused ischemic myocardium. In an atrial cardiac cell line, H9C2, the addition of SKF96365, another widely used inhibitor of TRPC, ameliorates injuries induced by hypoxia–reoxygenation (H/R) [24]. However, it is well known that 2-APB and La^3+^, as reviewed previously [75,76], as well as SKF96365 [75,76], are not specific to TRPC channels and may block other cationic channels. Therefore, these results should be supported by experiments using siRNA and/or TRPC-deficient mice. Actually, other reports used different molecular approaches to identify TRPC isoforms responsible for Ca^2+^ entry and its relationship with cardiac myocyte death under I/R. For example, Shan et al. [73] observed that transgenic mice overexpressing TRPC3 in myocardial cells are highly sensitive to injuries after I/R as they enhance apoptosis through increased TRPC3-mediated Ca^2+^ influx and calpain cleavage. They also demonstrated significant improvement in the viability of cardiomyocytes after SKF96365 treatment. Moreover, Meng et al. [77] observed that in vitro I/R increases TRPC6 protein expression, [Ca^2+^]_i_ levels, and cell apoptotic rate in a time-dependent manner in H9C2 cell line. In addition, they suggested TRPC6 as a possible target for cardioprotection in H9C2 cells since the administration of danshensu, an active component of *Salvia miltiorrhiza*, protects against I/R injury by reducing TRPC6 expression via the c-Jun N-terminal kinases (JNK) signaling pathway [77]. Hang et al. [78] also demonstrated that brain-derived neurotrophic factor (BDNF) protects against MI and inhibits H/R-mediated cardiomyocyte apoptosis through TRPC3 and TRPC6 regulation. 

On the other hand, the role of TRPC1 in I/R is still unclear. A recent study suggested that it is implicated in I/R injury, as the expressions of mRNA and protein of TRPC1, Orai1, and STIM1 are significantly increased in vivo in mice subjected to myocardial I/R injury and in vitro in H9C2 cells after H/R [79]. Interestingly, the suppression of STIM1 by siRNA decreases the expression of TRPC1 and Orai1, leading to decreased intracellular Ca^2+^ accumulation and apoptosis produced by H/R in H9C2 cells [79]. Therefore, STIM1 likely regulates the expression of TRPC1 and Orai1 in the context of apoptosis and myocardial I/R injury. In contrast, Al-Awar et al. [80] speculated that TRPC1 plays a cardioprotective role against I/R injury. They showed that sitagliptin, an inhibitor of dipeptidyl peptidase-4 (DPP-4), decreases the infarct size in a rat model of I/R which correlates with the increase in protein levels of TRPC1, TRPV1, and calcitonin gene-related peptide in heart tissue. Nevertheless, a specific experiment targeting TRPC1 was not shown. Our recent study, through Western blot, confirms that TRPC1 and 6 are upregulated in a rat model of I/R although they are not inhibited by urocortin-2-mediated cardioprotection. In contrast, urocortin-2 administration in NRVM undergoing in vitro I/R inhibits SOCE and prevents I/R-induced protein overexpression of TRPC5 and Orai1 [13]. Taking into consideration these results, further investigations are necessary to clarify the functional role of TRPC channel increase after I/R. 

## 6. TRPC Channels in Post-Ischemia Cardiac Repair 

After MI, the heart undergoes extensive adaptative processes and myocardial remodeling, involving angiogenesis, cardiac cell hypertrophy, and accumulation of fibrous tissue in both the infarcted and the non-infarcted myocardium, as reviewed previously [81,82,83]. Nonetheless, the role of the TRPC protein in cardiac repair still remains poorly studied.

### 6.1. TRPC Channels in Post-Ischemia Angiogenesis

Angiogenesis relies on new blood vessels forming from pre-existing vessels and the subsequent expansion of the vascular network in the body. Post-ischemic angiogenesis is considered a protective mechanism motivated by the lack of oxygen and blood supply necessary for physiological heart repair after a MI [84,85]. Angiogenesis involves sprouting, proliferation, migration, and tube formation thanks to the stimulation of endothelial cells (ECs) by growth factors such as vascular endothelial growth factor (VEGF), considered as the most potent pro-angiogenic factor specific for ECs (reviewed elsewhere [86,87]). Compelling evidence demonstrated that chronic and transient ischemia significantly increase the expression of VEGF [88,89,90]. Nevertheless, pre-clinical and clinical trials using solely pro-angiogenic factors, such as VEGF, were not shown to be effective in patients with stable angina or critical lower limb ischemia [91,92]. VEGF stimulates two tyrosine-kinase receptors, VEGFR-1 and VEGFR-2 [84,93], to increase [Ca^2+^]_i_ in ECs involving Ca^2+^ release from intracellular stores and extracellular Ca^2+^ flux through cation channels, such as TRP channels [87,94]. There is increasing interest in the role of TRPC channels in angiogenesis, especially in studies related to cancer and diabetes [95,96,97]. ECs express different TRPC proteins involved in vascular function (TRPC1, 4, and 6), vascular tone remodeling (TRPC4), and oxidative stress-induced responses (TRPC3 and 4) [98]. It is apparent that TRPC3 and 6 are implicated in VEGF-mediated [Ca^2+^]_i_ increase in ECs and angiogenesis. Indeed, VEGF- and OAG-induced extracellular signal-regulated kinases (ERK) 1/2 activation and tubulogenesis are significantly suppressed by TRPC3 inhibitor and siRNA in human umbilical vein ECs (HUVEC) [99]. Meanwhile, the overexpression of dn-TRPC6 in human microvascular ECs inhibits the VEGF-mediated [Ca^2+^]_i_ increase, migration, sprouting, and proliferation, well-known hallmarks of angiogenesis [100]. In addition, TRPC4 siRNA attenuates oxidized low-density lipoprotein (oxLDL)-induced human coronary EC proliferation, migration, and angiogenesis tube formation [101].

Unfortunately, little is known regarding the role of TRPC channels during post-ischemic angiogenesis. In contrast, TRPC channels appear involved in hypoxia-induced angiogenesis [96,102]. For instance, the expression of TRPC4 protein is significantly upregulated in the retina under hypoxic condition. TRPC4 siRNA inhibits VEGF-induced migration and tube formation of retinal microvascular ECs, which suggests a role of TRPC4 in initiating neovascularization in response to VEGF in retina under hypoxia [96]. Moccia et al. [103] hypothesized and debated whether transfecting TRPC3 into autologous endothelial progenitor cells (EPCs) might enhance revascularization and functional recovery of ischemic hearts. However, functional experiments that tested this hypothesis were not performed. Recently, Zhu et al. [104] demonstrated that TRPC5 activation is necessary for EC sprouting, angiogenesis, and blood perfusion in a hind-limb ischemia model. TRPC5 downregulation prevents NFAT activation and EC tube formation under hypoxia. Moreover, TRPC5 KO mice have worse vascular recovery than WT mice after an ischemic injury. Finally, activation of TRPC5 by riluzole stimulates ECs sprouting and significantly improves limb recovery from ischemia injuries [104]. Therefore, it will be interesting to confirm the beneficial role of other TRPC channels in post-ischemic heart angiogenesis. 

### 6.2. TRPC Channels in Early Adaptative Cardiac Remodeling

An early cardiac hypertrophy and fibrosis are considered compensatory events to the loss of cardiac myocytes, necessary for wound healing and scar formation after heart infarcts. However, prolonged hypertrophy could lead to the development of HF, arrhythmias, and even sudden cardiac death [105]. Since it is known that the activation of TRPC channels mediates the Ca^2+^ influx, which activates Ca^2+^ intracellular signaling pathways, such as calcineurin/NFAT, TRPC channels are suggested as Ca^2+^ effectors and transducers of hypertrophic genes in the heart. Little is known regarding TRPC channel implication in I/R-induced cardiac hypertrophy. In contrast, there is general agreement regarding the role of TRPC channels in pathological cardiac hypertrophy as a consequence of aortic constriction or under chronic GPCR stimulation using endothelin-1, phenylephrine, or angiotensin-II [31,106,107]. Similarly, Makarewich et al. [71] revealed an upregulation of TRPC1, 3, 4, and 6 channels in mice six weeks post MI as compared to sham animals, along with the activation of the so-called fetal gene program, commonly used as markers of cardiac hypertrophy. They also demonstrated that mice expressing dn-TRPC4 have less pathological hypertrophy, better cardiac hemodynamics performance, and increased survival after MI, as compared with wild-type (WT) mice, which all suggest a critical role of TRPC4 in post-MI heart damage. Cardiac hypertrophy is also observed in rat heart tissue as early as one week post I/R, which correlates with the upregulation of the expression of TRPC1, 3, 4, 5, and 6 mRNA [13] and the activation of the fetal gene program (unpublished data). Recently, Dragún et al. [108] examined the expression of TRP channels in 43 patients with end-stage HF. They discovered, among other TRP channels, a significant increase in TRPC1 and 5 gene expression, while TRPC4 expression was decreased in HF patients compared to a healthy donor. Also, they detected a significant correlation of the gene expression of TRPC1 and MEF2c (myocyte enhancer factor 2c), considered a key transcription factor for cardiac hypertrophy [109]. Interestingly, this pilot study did not detect any significant differences in TRP expression between male and female HF patients, nor between HF patients based on ischemic or non-ischemic background. Another recent study observed a similar increase in the expression of TRPC1 in hearts of patients with hypertrophic cardiomyopathy (HCM) or HF. This study further used human pluripotent stem cell lines of TRPC1 KO generated using clustered regularly interspaced short palindromic repeats (CRISPR)/ CRISPR-associated protein 9 (Cas9) to confirm the role of TRPC1 in regulating cardiac myocyte hypertrophy induced by phorbol 12-myristate 13-acetate (PMA), which was associated with abnormal activation of nuclear factor kappa-light-chain-enhancer of activated B cells (NF-κB) [110]. Altogether, this indicates that TRPC channels might play a similar role in cardiac hypertrophy and HF, independently of patient background. Once TRPC expression is triggered, they perhaps activate several Ca^2+^-dependent factors of transcription and cardiac hypertrophy genes, leading to the same outcome, i.e., HF.

In the case of fibrosis, multiple well-known markers of fibrosis, hypertrophy, and Ca^2+^ handling protein were identified recently using genome-wide transcriptome analysis of infarcted hearts [111]. TRPC6 is considered a regulator of myofibroblast differentiation, a hallmark of fibrosis, since its silencing in human cardiac fibroblasts attenuates the transforming growth factor beta 1 (TGF-β1)-induced upregulation of alpha smooth muscle actin (α-SMA), a marker of myofibroblast transformation [112]. A recent study confirmed that the serum level of TGF-β1 is increased 28 days after MI in mice, accelerating cardiac fibrosis [113]. On the other hand, Saliba et al. [114] described that polyphenol extracted from grape pomace decreases angiotensin-II-induced Ca^2+^ entry through a direct regulation of TRPC3 and subsequent activation of NFATc3 in human ventricular cardiac fibroblasts, which abrogates myofibroblast differentiation and fibrosis by decreasing collagen secretion. However, the direct contribution of TRPC channels in cardiac fibrosis mediated by ischemia was barely addressed. Different isoforms of TRPC proteins are upregulated in rats showing fibrosis one week post I/R, although their direct role in promoting fibrosis was not examined [13]. Interestingly, TRPC6, through calcineurin–NFAT signaling, seems to be required for myofibroblast transformation after MI, a critical step during which collagen deposition and scar formation happen to maintain ventricular wall structural integrity in the early days post MI. In fact, TRPC6 KO mice show poor wound healing and fewer myofibroblasts, stained with α-SMA antibody, in the infarcted area [37]. Moreover and independently of studies related to ischemia and heart infarct, several reports proposed the participation of TRPC channels in the cardiac interstitial fibrosis caused by pressure overload by thoracic aortic constriction (TAC) in animal models or using vasoactive agonists, such as phenylephrine [36,106,115]. For instance, experiments performed in TRPC1/4 double-KO mice revealed significant amelioration of pressure overload-induced hypertrophy and interstitial fibrosis, which is explained by a reduced activity of TRPC1- and 4-dependent basal Ca^2+^ entry in adult ventricular myocytes [65]. At the same time, TRPC3 knockdown, using a small hairpin RNA lentivirus through the tail vein of mice, efficiently suppresses the extent of atrial fibrosis induced by TAC [116]. 

## 7. Concluding Remarks

In light of the reviewed studies, TRPC proteins stand out as key ion channels critical for cardiac cell responses under ischemic stress. A clearly defined role for specific TRPC isoforms in cellular events related to ischemic heart diseases still remains elusive, perhaps reflecting the complexity of these channels, the limitations of pharmacological tools, and the lack of specific inhibitors and antibodies. Nevertheless, TRPC channels were extensively studied since they sense and respond to a plethora of endogenous and exogenous stimuli by Ca^2+^ signaling in cardiac cells. Increasing evidence indicates that TRPC channels contribute to pathophysiological consequences of heart infarction, such as cardiac hypertrophy, fibrosis, and post-ischemic angiogenesis, as summarized in Figure 1. The potential to influence these outcomes by specifically modulating the expression and/or function of TRPC channels requires major efforts and more investigation. Further progress in the mechanistic understanding of TRPC channels will certainly help to identify new therapeutic targets for drug development to mitigate the impact of ischemia on cardiac function and to prevent cardiac transition from adaptive responses to harmful heart failure.

## Figures and Tables

**Figure 1 cells-09-00173-f001:**
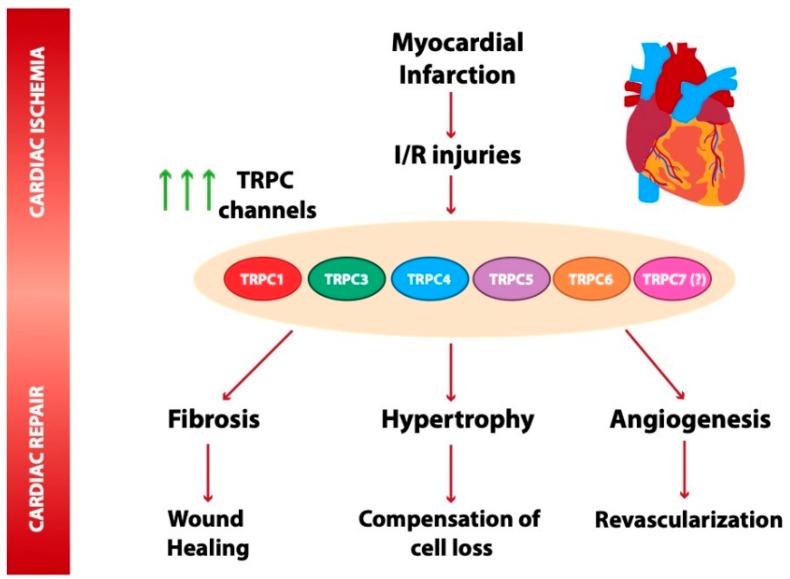
Scheme summarizing transient receptor potential canonical (TRPC) channel isoforms dysregulated under myocardial infarction (MI) and ischemia and reperfusion. TRPC1, 3, 4, 5, and 6 are upregulated in mouse and rat animal models of MI [5,13,71]. Compelling evidence indicates that TRPC channel overexpression contributes to Ca^2+^ entry, mediating the activation of Ca^2+^-sensitive signaling pathways, such as calcineurin–NFAT, a critical pathway involved in apoptosis, cardiac hypertrophy, and fibrosis [13,28,55,66,67]. TRPC proteins are likely also involved in cardiac repair-related processes. The protective role played by TRPC6 in wound healing is of note [37]. Other studies suggested a role of TRPC channels, such as TRPC5, in angiogenesis and revascularization triggered post ischemia [104].

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
