# Peer review of "TRPC Channels: Dysregulation and Ca2+ Mishandling in Ischemic Heart Disease"

_cells, 2020, doi:10.3390/cells9010173_

Round 1

Reviewer 1 Report

In the manuscript “TRPC channels: Dysregulation and Ca2+Mishandling In Ischemic Heart Disease” D. Falcón and coworkers made a review of the literature regarding the evidence and/or proposed role of TRPC channels in cardiac disease with focus on cardiac ischemia situations. They start with an introduction regarding the excitation-contraction coupling in the heart and generalities of the TRP ion channel family. After the introduction (1) there are 5 sections: 2) TRPC channels in the cardiovascular system, includes the expression of TRPCs in the heart with examples  in different cell types; 3) TRPC channels mediated Ca2+influx in cardiac myocytes, basically restricted to TRPCs and store-operated Ca2+ entry (SOCE) in cardiomyocytes; 4) Role of TRPC channels in cardiac physiopathology, short revision mentioning that TRPCs have almost no role under physiological cardiac conditions; 5) Role of TRPC in cardiac ischemia, here the topics are (5.1) TRPC in myocardial infarction and (5.2) TRPC role in ischemia and reperfusion injuries, both subsections accompanied from several examples found in the literature; 6) the follows a section about TRPC in post-ischemia cardiac repair, subdivided into (6.1) TRPC in post-ischemia angiogenesis and (6.2) TRPC channels in early adaptative cardiac remodeling, also both sections mention several publications associated with the topics. Finally, the conclusions pointed out the requirement of more research on this field and it is accompanied with a figure linking TRPC expression and cardiac pathophysiological processes linked to myocardial infarction.

Major remarks

One of the main purposes from a review should be to integrate and critically summarize available information regarding a specific topic of interest. This includes presenting all evidence that supports or opposes to a determined hypothesis as neutral as possible, and if it is discussed, it is also require tpo present the advantages and disadvantages of the approaches (blockers, specificity of the antibodies used). This review in some part lacks what above mentioned and it gives the impression that they authors want to present mainly evidence supporting own recent work from this group (Ref#7), but omitting important publications on the field of TRPC and cardiac disease.

For example in section 3, the revision on TRPCs and Receptor Operated Ca2+Entry (ROCE) pathway in cardiac cells is omitted and focused on SOCE which is a protocol to evoked Ca2+ entry by artificial blockage of SERCA.  What about the effect of AngII, one of the main cardiovascular remodeling agents, on TRPCs? If the intention of the review is to motivate the study of physiology.

There are statements (see below) presented along the review supported by publications were the information mentioned is not found or mentioned. These kinds of statements are dangerous, especially for those readers who are not deeply familiar with the topic, they will take and distribute information that is not completely correct., e.g lane 172/173 the role of TRPCs after I/R is highlighted , but in the papers cited (54-56) the role of TRPCs is not studied!

Other issues.

Line 102: has not been clarified yet (“ ..completely clarified” seems more appropriate). Lines 118-119: Conclusion about composition of TRPC tetramers is rather speculative from the references mentioned. Heteromultimerisation has not been convincingly studied in cardiac cells. Lines 121-24: In the way is presented the information one can assume that if the TRPC3 transgenic have more contractility the KO should have less (?). Overexpression of proteins can disrupt the natural interaction partners just by changing the stoichiometry of the available protein. Line 128: Mice instead of mouse Lines 128-129: “Any abnormalities” All global TRPC-KO mice are available and survive, even the hepta-KO (PMID: 26377676). However, studies of the cardiac functions mentioned in TRPC single and compuond TRPC KO mice should be reviewed by the authors, e.g. herat rate is changed in TRPC1/TRPC4 Ko mice (PMID: 26069213) ; how about the other published mouse lines? Anrep effect depends on TRPC3/TRPC6 (PMID: 24449818). Line 142: Add after TRPC either “channels” or “proteins” (apply also to other subheadings). Subsection 5.1: One important publication with high impact on the field analyzing the role of TRPC6 after MI is omitted (Davis et al., 2012. Developmental Cell. 23: 705, PMID: 23022034). This paper shows using TRPC6-KO mice results in the same direction as those reviewed in lines 268-272, but already before and also can be there discussed. section 3, Page 3 about the role of TRPCs for SOCE: The authors should state that despite numerous studies about the role of TRPC in SOCE in cardiomyocytes, this concept has never been confirmed by the appropriate control , i.e. evaluation of SOCE in cardiomyocytes lacking expression of the corresponding TRPC protein. Line 177: Please add the reference number after He et al. because the references are not alphabetically organized and can be misleading. This applies several times along the text. Line 181: Here one can mention also TRPC-deficient mice since there are many studies in the field with antibodies that do not specifically recognize the TRPC protein against which the antiserum was raised. Line 192: Define BDNF. Line 197: Check grammar. Line 203: Please mention in which cells it was observed. This would be desirable for several examples where it is no clear which cells were used and it would help the reader to understand the role of the TRPCs on specific cell types beyond just the “heart”. Line 208: Please specify TRPC increase on what level (mRNA, protein,…)? Line 218: Omit “,” after migration (look this use of comma along the text) Line 226: Change “interest on” instead of “interest of” Line 226-227: The authors highlight studies about angiogenesis in cancer; what then about pathological processes like Diabetes? Here also angiogenesis plays a dominant role of diabetic complications and TRPCs are involved. This is motivated in part because in reference 83 a Hind-limb ischemia model and the retinal angiogenesis analysis were presented in TRPC5-KO and others also analyzed i.e. angiogenesis during diabetes development in TRPC-deficient mice. Line 247: Add “the” before Ca2+. Line 252: “there is a general agreement regarding the role of TRPC3 and TRPC6 in pathological cardiac hypertrophy” What about other studies about the role of TRPC1, TRPC4, TRPC1/C4 in this process? There are still controversies and mainly coming from strong models like TRPC-deficient mice. That the amount of publications is larger on one direction does not imply a general agreement; findings need to be discussed based on facts and not on common agreements. Line 262: “Similarly” seems not appropriate. Line 263: Check commas Lines 266-268: “…there are several studies that proposed the participation of TRPC3 and TRPC6 in interstitial fibrosis caused by thoracic aortic constriction in animal model or using vasoactive agonists as phenylephrine [25,94,95].” This affirmation only can be applied to reference 95. Line 274: Please elude in more detail what is meant by “fibroblasts phenotype” Lines 293-295: Somehow confusing here, on one side the argumentation under “Concluding remarks” in the text is that the role of TRPC proteins in ischemic cardiac disease “…stills remain elusive” and in the figure legend is written “…studies demonstrated the role of TRPC in post-ischemic cardiac repair…” A more balanced statement about which processes critically depend on TRPC and for which the evidence is still weak and needs confirmation would be very beneficial for this review here as well as at multiple other positions throughout the manuscript. line 103: DAG can activate also TRPC4 and TRP5 (see Storch, et al, …, Gudermann, PNAS 2017, PMID: 27994151).

Author Response

Thank you very much for your deep revision and recommendations. We sincerely apology for all the mistakes found in the manuscript. It wasn´t our mean to omit any important publications of TRPC in cardiac disease. Looking at this list of authors invited to this topic we hint that others will address this issue TRPC role in different cardiovascular diseases. Our major objective was to focus on heart infarction and ischemia-related disease. For this reason, the first sections of this manuscript were planned to give a brief general overview of TRPC expression, role in Ca2+ entry and cardiac physiopathology. Definitely, we didn’t want to discard any important TRPC works, neither to focus only in our previously paper. Sorry if we gave this impression

We followed carefully, point-by-point, all your recommendations and performed substantial changes as highlighted in this revised manuscript. We attached a word document detailing all our answers. 

In summary:

We added a new paragraph discussing the concept of heteromultimerization of TRPC in cardiac cells. Line 96-103. We added and discussed the role of TRPC in ROCE signaling pathway in cardiac cells. Line 118-130. We revised and discussed the participation of TRPC in SOCE in cardiac cells. Line 152-157 We revised and added more information in section 5 We revised all the citations and change those added by mistakes. For example in pag 6 line 219.

Finally, we are aware that some issues remain unclear and might require more detailed revision. We honestly appreciate your efforts in improving this manuscript and we hope that you will find this improved version suitable for its publication.

Reviewer 2 Report

The authors have given a summary of most recent literature on TRPC in cardiac ischemia/MI. While I agree with the fact that the manuscript is of potential interest to the academics and students in the specified field, I have following concerns.

Language requires further attention and careful evaluation; grammatical errors checks and reformatting of unnecessary compounded sentences will greatly enhance the impact of the review. Organisation of the facts is poor; authors have to work on flow of the text and connectivity between sentences/paragraphs. Complete duplication of sentences - few sentences were repeated word by word in the manuscript's different parts. The manuscript stands as the summary of the recent literature-rather than critical evaluation of them. At this level an attempt should be made to evaluate and highlight the conflicting literature and then give a reasonable conclusion with a direction for further research questions.

Author Response

Thank you very much for your revision and recommendations. We have made substantial revision through the whole manuscript and reorganize it for better understanding. We highlighted and discussed different concept regarding the role of TRPCs in cardiac disease related to heart infarction. In summary:

We added and discussed the concept of heteromultimerization of TRPC in cardiac cells. Line 96-103. We added and discussed the role of TRPC in ROCE signaling pathway in cardiac cells Line 118-130. We revised and discussed the participation of TRPC in SOCE in cardiac cells We added a paragraph describing and discussing the effect of TRPC on cardiac contractility. We revised the specific implication of those TRPCs dysregulated under I/R and heart infarct.

We hope that this valuable reviewer finds this revised manuscript suitable for its publication.

Round 2

Reviewer 1 Report

I only hew have the following issue:

Line 103 and 110 (hippocampus), heteromultimerissation: please add the recent study in the reference list (PMID: 28790178) that used quantitative high-resolution mass spectrometry and protein fractions from corresponding KO controls to demonstrate that in mouse brain and hippocampus TRPC1, TRPC4, and TRPC5 assemble into heteromultimers with each other, but not with other TRP family members. 

This study was performed in mouse brain and hippocampus, but  this tissue was mentioned/cited also in ref 44, and this recent study uses now a quantitative and more comprehensive approach and, in addition, uses controls from knockout tissues.

Author Response

The reference required by the reviewer (PMID: 28790178) has been already included in the original version of this manuscript as ref [43]. Now, we comments finding of this study for better understanding.